# Small Felids Coexist in Mixed-Use Landscape in the Bolivian Amazon

**DOI:** 10.3390/ani14050697

**Published:** 2024-02-23

**Authors:** Courtney Anderson, Amelia Zuckerwise, Robert B. Wallace, Guido Ayala, Maria Viscarra, Oswald J. Schmitz

**Affiliations:** 1School of Environment and Natural Resources, The Ohio State University, Columbus, OH 43210, USA; 2School for Environment and Sustainability, University of Michigan, Ann Arbor, MI 48109, USA; ameliaz@umich.edu; 3Wildlife Conservation Society, La Paz, Bolivia; rwallace@wcs.org (R.B.W.); gayala@wcs.org (G.A.); mviscarra@wcs.org (M.V.); 4School of the Environment, Yale University, New Haven, CT 06511, USA; oswald.schmitz@yale.edu

**Keywords:** small cats, conservation, occupancy modeling, human activities, carnivore guild, biodiversity, ocelot (*Leopardus pardalis*), jaguarundi (*Herpailurus yagouaroundi*), margay (*Leopardus wiedii*), oncilla (*Leopardus tigrinus*)

## Abstract

**Simple Summary:**

Humans are impacting the planet like never before. In the face of global species loss, many have concluded that the only way to protect species is for humans to remove themselves or be forcefully removed from landscapes. Our study investigates four species of small wild cats in the Bolivian Amazon to determine if they are negatively impacted by human activity, such that they are less likely to occur near human settlements or tourism centers. Field camera data reveal that at least two of these species do not appear to avoid human activity centers, including tourist lodges and indigenous communities, suggesting that these do not conflict with the wild cats’ needs in this landscape. We did not observe negative effects between species that would indicate competition, which suggests high habitat quality or the effective division of roles amongst the cats. Our results illustrate that the combined efforts of government and indigenous peoples in the Greater Madidi Landscape have been successful in protecting these vulnerable species.

**Abstract:**

In the face of global species loss, it is paramount to understand the effects of human activity on vulnerable species, particularly in highly diverse, complex systems. The Greater Madidi Landscape in the Bolivian Amazon includes several biodiverse protected areas that were created with the goal of sustaining healthy and diverse ecosystems while not impeding the livelihoods of local indigenous peoples. In this study, we sought to use camera trap data and single-species occupancy analysis to assess the impacts of different forms of human activity on four species of small felids: ocelots (*Leopardus pardalis*), margays (*Leopardus wiedii*), jaguarundi (*Herpailurus yagouaroundi*), and oncilla (*Leopardus tigrinus*). We modeled both human variables (proximity to indigenous communities, roads, and tourist camps) and non-human variables (terrain ruggedness, proximity to rivers, canopy height, prey availability, and large cat abundance). Margay occupancy was unaffected by any of these human variables and ocelots showed only weak evidence of being affected by tourism. Ocelots were particularly pervasive throughout the study area and were consistently estimated to have high occupancy probability. We did not obtain sufficient data on jaguarundi or oncilla to reliably model these effects. Our results indicate that small cats successfully coexist both with each other and with the surrounding human activity in this unique landscape, which serves as a model for global protected area management.

## 1. Introduction

Human influence on Earth’s ecosystems is widespread and growing, particularly in tropical areas that are rich in biodiversity [1,2,3]. Land use change has left as little as 25% of terrestrial landscapes in an unaltered state [3], precipitating an accelerated loss of biodiversity [3,4,5,6]. But is the concern that human presence is always detrimental to ecosystems and biodiversity well founded? Not all protected areas are free from pressures of human land use [7], and not all human activity is equally destructive to natural systems [8,9]. With many species experiencing differential responses to human activities, and the nature of human activity varying widely, it becomes necessary to consider species-specific impacts of locally relevant anthropogenic pressures.

Marneweck et al. [10] hold that small carnivores are ideal indicators of environmental change because of their sensitivity to localized impacts. To this end, we focus on a guild of four small cat species that are functionally diverse in terms of impacting ecosystem functioning. Functional diversity is determined by variation in habitat use and the hunting style of predators [11]. The four cat species—ocelot (*Leopardus pardalis*), margay (*Leopardus wiedii*), jaguarundi (*Herpailurus yagouaroundi*), and oncilla (*Leopardus tigrinus*, also called “northern little spotted cats”)—exhibit morphological and behavioral diversity (Figure 1). As such, despite a lack of knowledge of hunting behavior, these species are thought to fill a variety of niches based on habitat use and prey utilization [12,13]. Margays, and to some extent oncillas, are arboreal predators, whereas ocelots and jaguarundis are primarily ground-level hunters [14,15,16,17]. The three smallest cats (oncillas, margays, and jaguarundis) prey predominantly on small animals (<1 kg) [14,18,19,20,21]. Jaguarundis and margays, while closest in size and hence prey size utilization, minimize spatial overlap via habitat choice and minimize temporal overlap with jaguarundis being diurnal and margays being nocturnal [14,22,23]. Ocelots have more generalist diets that include larger prey (<15 kg) [14,18,19,22,24,25]. All four species have relatively small home ranges, and relatively short life histories. Together, these attributes of this small cat guild provide a high degree of sensitivity to local land use changes, making them ideal sentinels to the effects of human alteration of the forest ecosystem [10]. Moreover, these cat species are among the one-third of all small carnivore species that are considered Threatened or Near Threatened with extinction by the IUCN Red List and the one-half of all small carnivores that are in decline [26], making their monitoring all the more important. Research has shown that these species respond to human influences in different ways [27,28]. With the potential for these animals to serve as indicator species, and their conservation concern as species in decline, it is important to understand their current habitat occupancy and the effects of human activity on each species.

We therefore conducted a camera trap survey in the Greater Madidi Landscape in the Bolivian Amazon, within two national protected areas: the Madidi National Park and Natural Integrated Management Area and the Pilón Lajas Biosphere Reserve and Indigenous Territory. The Greater Madidi Landscape is a well-studied region of considerable biodiversity [29]. Sites were chosen based on different types of human activity, including areas of residence of indigenous peoples, areas of tourism, and a relatively secluded area with little human influence (a quasi-control for human impacts). We then investigated the occupancy probability of the four small cat species in relation to proximity to human activity centers (e.g., towns, tourist lodges). The small cat guild co-occurs with larger cats—jaguars (*Panthera onca*) and pumas (*Puma concolor*). Jaguars and pumas can exert strong competitive and/or intraguild-killing behaviors toward smaller carnivores [30]. We accounted for this potential effect on small cat occupancy using jaguar and puma relative abundance as covariates. Additionally, previous evidence also suggests an “ocelot effect” [14,31] involving the suppression of jaguarundis, margays, and oncillas by the more dominant ocelots.

Here we report a study that explores the impacts of different human activities on four small cat species in a unique landscape of overlapping indigenous territories and national protected areas. We hypothesized the following:(1)Occupancy probabilities will decrease with increased proximity to human activity centers.(2)Occupancy probabilities will be lower at stations with higher jaguar or puma relative abundances. Smaller cat (oncilla, jaguarundi, and margay) occupancy probabilities will be lower at stations with higher ocelot occupancy estimates.

We predicted that the smaller three species would be less sensitive to these human-proximity effects than ocelots, due to their larger home ranges, lower energetic demands, and potential for decreased competition if ocelots are repelled by human areas. We also predicted that human activity would have a stronger effect on occupancy probability than prey or environmental factors.

## 2. Materials and Methods

### 2.1. Study Site

Data were collected from three sites within the Madidi National Park and Natural Integrated Management Area and the adjacent Pilón Lajas Biosphere Reserve and Indigenous Territory in the La Paz Department of Bolivia, located in the Andean foothills at approximately S 14°37′30.0″, W 67°43′06.4″ [32]. The area is composed of a tall, evergreen forest in the Amazonian-pre-Andean seasonal forest zone [32], which receives an annual mean precipitation of 2440 mm and mean annual temperature of 24.9 °C [33]. The Greater Madidi Landscape is divided into management zones with distinct regulations for permitted activities. Our study was conducted between June and November 2019, which corresponds with the region’s dry season [34,35], as the failure rate of the cameras during the rainy season becomes prohibitive. This study period was divided into three surveys, each along a different river that experiences distinct human pressures (Figure 2).

The first survey was conducted along the Quiquibey River in the Pilón Lajas Biosphere Reserve, where members of the Tsimane-Mosetene indigenous community reside, hunt, grow small-scale crops, and collect forest products [36]. The following two surveys were conducted along rivers within the Madidi Protected Area, specifically in the Madidi Integrated Management Natural Area, where tourism and regulated non-extractive forest use is permitted [37]. The second survey was specifically conducted along the Hondo River, a smaller tributary that is poorly navigable during the dry season and effectively restricts most access by boat. This site overlaps the Lecos Apolo indigenous territory. The third survey was conducted along the Tuichi River, which is close to the town of Rurrenabaque and the site of most of the protected area’s ecotourism, and which overlaps with the Tacana, San José de Uchupiamonas, and Lecos Apolo indigenous territories. Data from the three surveys were pooled under the assumption that the study species did not alter their occupancy/habitat use within a single season.

### 2.2. Camera Trap Surveys

Data were collected using motion-activated camera trap stations, where each station consisted of two independent motion-detecting trail cameras (Reconyx HyperFire 2), set approximately 3 m apart and facing toward a central location cleared of debris [35,38,39,40,41]. Specific camera settings and protocols can be found in Appendix A. A scent lure was placed between the cameras at each station to increase detection and entice animals to remain in front of the cameras for better identification [41,42,43]. Studies have shown a scent preference by jaguars for both Calvin Klein Obsession for Men [44,45,46] and Chanel No. 5 brand colognes [45,46]. Since these scents were untested on the small felid species, we systematically alternated between the two at each station.

Stations were set in a grid with cells of 1 km^2^, with an average of 1014 m between stations. This spacing is intended to maximize the detection of the small felid species, such that none of the species’ home ranges would fall entirely within the space between camera stations [40]. Home range estimates from previous studies suggest a 2 km^2^ female ocelot minimum home range area in the Amazon Basin [25,35]. Margays, being small-bodied and semi-arboreal, are understudied and the few studies estimating their home ranges have shown high variation [47,48,49]. Harmsen et al. [50] recommend spacing cameras no greater than 1 km apart for the optimal capture of margays. Similarly elusive, jaguarundi home ranges have shown even more variation [15], though Michalski et al. [51] estimated a female home range at 1.88 km^2^. Oncillas are one of the most poorly studied felids, and data on their space use are very lacking [52,53]. Reported home range sizes vary from 0.9 to 17 km^2^ [14,52]. Though research on these species has shown high variation in home range sizes, it is highly dependent on habitat and prey availability [22,54]. In a previous study on ocelot space use in the Madidi area, Ayala and Viscarra [35] found particularly high densities, with smaller than average home ranges. For this study, we assumed that our camera spacing was appropriate for occupancy estimation for all four of these species, with the understanding that oncillas, margays, and jaguarundis are potentially underestimated.

Within each 1 km^2^ grid cell, hypothetical point locations were placed randomly, while keeping the minimum distance between stations at roughly 700 m, to reduce bias [40]. The camera stations were installed within 100 m of this hypothetical point in a location considered likely to optimize detectability, such as along animal trails or near fresh signs of a focal species or known prey [35,40,44].

A total of 149 camera stations were deployed throughout the study area, with 59, 52, and 38 stations in the first, second, and third surveys, respectively (Figure 2). All surveys were run for 46 days (following [34,35,41]) for a total of 6854 camera trap days.

### 2.3. Data Analysis

Photos were manually inspected and tagged with the animals in each frame identified to species level (except for small rodents, marsupials, and reptiles), including the target species and other “bycatch” species. Observations were considered independent events if (1) successive captures were of different species or (2) successive captures of the same species were separated by at least 30 min [35,40,55]. These independent observations were used to generate detection histories for the four small felids.

Independent observations of jaguars, pumas, and groups of prey species were used to calculate relative abundance indices (number of observations/100 trap nights) which were included as model covariates [40,55,56]. Relative abundance indices were used to approximate activity levels of jaguars, pumas, and prey species at different camera stations due to their home ranges varying substantially from the 1 km^2^ cell size, which would lead to over- or underestimation using an occupancy framework [57,58]. Based on studies of each species’ preferred diet, oncilla, margay, and jaguarundi prey were considered to be any animal <1 kg [14,18,19,20,21]. It is known that smaller animals are not as frequently detected on camera traps [38], so this prey relative abundance estimate is likely an underestimate. Ocelots have been shown to eat a higher proportion of moderately sized prey (<15 kg) [18,25,59,60]. A full list of prey species included for each felid is included Appendix B.

We then characterized both human and non-human predictor variables that we hypothesized would affect small felid occupancy (Table 1; Appendix C). Human activity variables included distance to the nearest road, distance to the nearest town, and distance to the nearest tourist lodge/camp. In addition to the relative abundance of species-specific prey, jaguars, and pumas at each site, non-human variables included the distance to the nearest river, mean terrain ruggedness index (TRI) [61], and canopy height. The scale and resolution of these covariates are intended to represent the landscape at the 1 km^2^ cell size, rather than the microhabitat features at each camera site. All covariates were scaled and centered prior to being used in occupancy models.

To model the impact of human activity on small felid presence, we applied occupancy modeling in a maximum likelihood framework. This defines occupancy probability (Ψ) as the probability of a species’ presence at a site, while the detection probability (*p*) is the probability that a species will be detected at a site, given that it is present [57]. Species’ habitat use can thereby be estimated, given that the species is not guaranteed to be detected even when present, a critical assumption when working with rare and elusive species such as small cats. Covariates can then be modeled that may differentially affect species’ occupancy probability or species’ detection probability.

We analyzed our data in single-season, single-species occupancy models, which assume that the occupancy state did not change throughout the study period [62]. This analysis was performed in R [63], using the package “RPresence” [64]. Each survey was analyzed as if they were performed concurrently, with survey ID (1–3) included as a model covariate. After testing a null model with different numbers of sampling periods for model optimization, we divided the survey period into four sampling occasions [24,65]. A global model of all covariates was run for each species. Ocelot occupancy probability at each station was included as a covariate in the margay, jaguarundi, and oncilla occupancy models. Scent lure type was incorporated as a detection covariate. As such, the model for ocelots was the following:Ψ(Survey + Dist. Town + Dist. Tourism + Dist. River + Prey Rel. Abund. + Mean TRI + Canopy Height + Jaguar Rel. Abund. + Puma Rel. Abund.), *p*(Scent Lure)(1)
and the model for jaguarundis, margays, and oncillas was the following:Ψ(Survey + Dist. Town + Dist. Tourism + Dist. River + Prey Rel. Abund. + Mean TRI + Canopy Height + Jaguar Rel. Abund. + Puma Rel. Abund. + Ocelot Occupancy Estimate), *p*(Scent Lure).(2)

Models were assessed for goodness-of-fit using 10,000 bootstrap iterations, following MacKenzie and Bailey [65]. If a lack of independence (overdispersion) was identified, standard errors of β-coefficients were adjusted by dividing them by the overdispersion parameter ($\hat{c}$). Variable significance was assessed using z-tests (Appendix C).

## 3. Results

A total of 44 species were identified to a species level. We recorded 478, 47, and 19 independent camera captures of ocelots, margays, and jaguarundis, respectively. We only obtained five records of oncillas, at two adjacent camera stations, so this species was excluded from the analysis. The goodness-of-fit test showed a slight overdispersion ($\hat{c}\approx 1.5$, *p* = 0.10) in the ocelot model and significant overdispersion ($\hat{c}>2$, *p* = 0.0068) in the jaguarundi model. The overdispersion adjustment was therefore only made for jaguarundi, though this adjustment was insufficient to generate reasonably sized coefficients.

Ocelots were detected at 72.5% (108/149) of camera stations, with the highest percentage of stations with detections in the third survey (Table 2). There were 215 (45.0%), 186 (38.9%), and 77 (16.1%) independent ocelot observations in the first, second, and third surveys, respectively. The ocelot occupancy probability (mean $\hat{\Psi }$ ± SE = 0.7760 ± 0.1274) was significantly higher in the third survey (Figure 3A and Figure 4A). The ocelot occupancy probability was not significantly related to any of the environmental or prey variables, nor was it significantly related to the distance to the nearest town (Figure 4A). Ocelot occupancy was positively correlated with the distance to tourist areas (p=0.0214), suggesting some avoidance of these tourist sites (Figure 5). This finding supports our hypothesis that occupancy would be lower near human activity centers and our prediction that occupancy would be more strongly impacted by human-related variables than environmental or prey-related variables. There was no evidence of the suppression of ocelots by larger species. The detection probability for ocelots was 0.4711 (SE = 0.02753). The detection coefficient for the different scent lures was very close to zero and ocelot observations were relatively even between stations with each lure (241 Calvin Klein, 234 Chanel), suggesting no preference between scents by ocelots.

Margays were detected at 18.8% (28/149) of camera stations, with higher percentages of camera stations with detections in the second and third surveys. There were 14 (29.8%), 18 (38.3%), and 15 (31.9%) independent margay observations in the first, second, and third surveys, respectively. The margay occupancy probability (mean $\hat{\Psi }$ ± SE = 0.3894 ± 0.1625) was not significantly different between the three surveys (Figure 3B and Figure 4B). The margay occupancy probability was significantly positively correlated with mean TRI (p=0.0394) and the distance to the nearest river (p=0.0151). Contrary to our hypothesis that human activity centers would negatively impact occupancy, none of the human-related variables in our model were significant predictors of margay occupancy probability. The positive relationship with the distance to the river indicates a higher occupancy probability at stations further from rivers. It is possible that this is a way of avoiding human activity since the rivers of Madidi are the main means of transportation around the region. If this were the case, we could expect that margay occupancy estimates would be less correlated with the distance to the river at stations from the second survey, which was conducted along a river with very restricted human access. As seen in Figure 6A, the correlation remains in all three surveys, which indicates a habitat preference to be further from rivers for some reason not related to human activity (e.g., preferred prey or habitat structure). Likewise, we divided the occupancy probabilities by survey when plotting mean TRI, as seen in Figure 6B. This was to visualize any potential confounding effects of the different sites. The correlation does appear to be stronger in the second and third survey sites, with higher estimates of margay occupancy at stations with an above average mean TRI, but the trend is still discernible at the first survey site.

Contrary to our hypothesis of the suppression of margays by larger cats, including the ocelot, we found no evidence for any relationship between the relative abundance of jaguars or pumas, nor ocelot occupancy estimate, on the occupancy probability of margays. The detection probability for margays was 0.1466 (SE = 0.0310), which is less than a third of the ocelot detection probability. The detection coefficient for the scent lure used was very near zero and non-significant, and margays were detected at 29 and 18 stations with Calvin Klein and Chanel brand lures, respectively, showing no obvious preference.

Jaguarundis were detected at 8.7% (13/149) of camera stations, with the highest percentage of camera stations with detections in the first survey. There were 12 (63.2%), 5 (26.3%), and 2 (10.5%) independent jaguarundi observations in the first, second, and third surveys, respectively. Despite the model reporting convergence, the estimated coefficients and standard errors were unreasonably large, suggesting that the model had insufficient detections for reliable estimation. Jaguarundis were detected at 11 and 8 camera stations with Calvin Klein and Chanel brand lures, respectively, showing no obvious preference.

The full list of coefficients from each species’ occupancy model is given in Table A1 Appendix C.

## 4. Discussion

We found only weak evidence of small felid occupancy being affected by human activity. We found no support for the intraguild suppression of smaller species by larger cats. In agreement with previous studies [27,28], different species show distinct responses to certain variables of human presence. While our study period was restricted to the region’s dry season, and our results are therefore only representative of that season, it is important to note that this is also the time of year when the area experiences the highest ecotourism activity.

The ocelot occupancy probability was not predicted by any of the environmental or prey availability covariates in our model. The camera stations from the third survey had significantly higher occupancy probability estimates than those from the previous surveys, but this was either an artifact of the temporal difference (each successive survey was further along into the dry season) or due to an unmeasured variable. This is supported by the slight overdispersion in this model, which indicates variation not accounted for by our model. Our covariates were calculated to model environmental effects at the cell level, so it is also possible that these variables have scale-dependent effects that were not captured at this resolution. The site sampled during this third survey was along the Tuichi River, which receives the most tourism activity of the sampled areas. Also, the distance to the nearest tourist lodge was the only other significant predictor, suggesting that, though there is potentially a higher occupancy probability in this region, on a more local scale, the probability is lower when approaching the tourist sites themselves (Figure 5). This result partially supports our prediction that human variables would be a stronger driver than environmental/interspecific variables, though the distance to the nearest indigenous towns was not significant.

The margay occupancy probability was significantly correlated with only environmental variables: distance to the nearest river and mean TRI. This is in opposition of our prediction that human factors would exert a stronger influence than environmental/interspecific factors, supported by the fact that margay occupancy was predicted to be higher with a greater distance from rivers in all three surveys, including the second survey which was conducted in an area with very low human traffic. The data did support our prediction that the smaller cats would experience weaker human-suppression effects than ocelots, which did show a significant relationship with tourist lodges.

The jaguarundi model did not identify any significant covariates. The large size of the estimated β-coefficients and related standard errors suggest that, though the model converged and was adjusted for overdispersion, the resulting model is not a good fit for the data. It is likely that jaguarundis were not detected enough times or at enough stations for the model to accurately estimate occupancy. Alternatively, jaguarundis may base their habitat use on some unmeasured variable. Little is known about jaguarundi ecology, making it especially difficult to predict which habitat/interspecies variables might be most significant. Studies have shown that jaguarundis feed on a large proportion of birds and reptiles [15,22,31], which are not always discernible in camera trap photos, making our prey relative abundance estimate a potential underestimate for this species and therefore less likely to predict occupancy.

Only five observations of oncillas were recorded, so we were not able to include them in the analysis. This is not surprising considering that they are known to occur in very low densities, especially in areas with high densities of ocelots [56]. The stations that detected oncillas were from the first survey, which was conducted in the area with indigenous communities, suggesting a tolerance of human activity. Further research is needed to understand oncilla ecology and their sensitivity to human activity.

There was no evidence of the suppression of small cat occupancy by larger cats. The ocelot occupancy probability was not influenced by the relative abundance of jaguars or pumas, and margay occupancy was not influenced by jaguar or puma relative abundance or the ocelot occupancy estimate. These results are in contrast to studies that have shown a distinct suppression effect of ocelots on smaller felids (the “ocelot effect”) [14,31] and suggest that the species of this guild may be coexisting via effective niche partitioning and responding to other drivers of habitat use in this study area. Alternatively, it could be that ocelots are so pervasive in Madidi that there is not enough variation in ocelot occupancy to detect potential responses by the smaller cats. Margays are also the most arboreal of this guild, which may increase their capacity to coexist with ocelots but also likely reduces their detection probabilities from ground-level camera traps. Our results may be less indicative of margay habitat use overall, but rather of margay ground use, in relation to our covariates. It may be that jaguarundis and oncillas would show stronger effects of suppression by ocelots, but we were not able to reliably model occupancy for those species. Given the low detection rates of these naturally rare small cat species, multispecies occupancy modeling approaches that could more directly evaluate species interactions/competition were not possible. Further study is needed to better understand the mechanisms of coexistence between these sympatric felids.

The only significant human-related variable from our models was the distance to the nearest tourist area, and this was only for ocelots. The fact that ocelots also had significantly higher occupancy estimates during the survey near the tourism area emphasizes that this effect is very localized. All three small cat species modeled had the greatest number of observations during the first survey (and the only survey to detect oncillas), which was also the region that had the highest level of human activity.

These findings are meaningful in the face of global pressures to increase species conservation by creating peopleless “protected areas.” Biodiversity loss is a planetary crisis, but initiatives to delineate a percentage of the Earth to be set aside from any human influence are problematic [66,67,68]. Not only do such land-sparing projects often come with serious social justice consequences [9,68], but they are not always effective [7]. We stress that our results are not representative of all human activity but that they instead reflect the success of the collaborative conservation efforts between Bolivian National Parks and the local indigenous peoples. Forest loss due to logging operations caused alarm that led to the park’s creation in 1995, an initiative largely driven and supported by the indigenous communities [69]. Ongoing conservation and research throughout the Greater Madidi Landscape has benefited from this indigenous support, and species have subsequently recovered [34,35,55]. Our results support previous work showing that this integration of indigenous-led and governmental efforts appears to have led to effective management of the Greater Madidi Landscape and its species [70,71].

## 5. Conclusions

Our results show a surprising lack of effect of human activity on the occupancy probabilities of three small felids—ocelots, margays, and jaguarundis. Ocelots appear to be somewhat impacted by tourist lodges but only at a very local scale. Margays and jaguarundis showed no correlation with any of our human activity variables, but margays were influenced by two non-human environmental variables (distance to the nearest river and mean TRI). None of these three cats showed suppression by jaguars or pumas, and margays and jaguarundis did not appear suppressed by ocelots, though the jaguarundi model is not reliable, likely due to very low detection rates. Since the majority of jaguarundi observations (12/19) and all oncilla observations were captured in the first survey, which had the most camera stations, it is possible that a more extensive camera array is necessary to adequately observe these low-density species. Overall, these data suggest not only coexistence between species but with the local human land uses as well.

## Figures and Tables

**Figure 1 animals-14-00697-f001:**
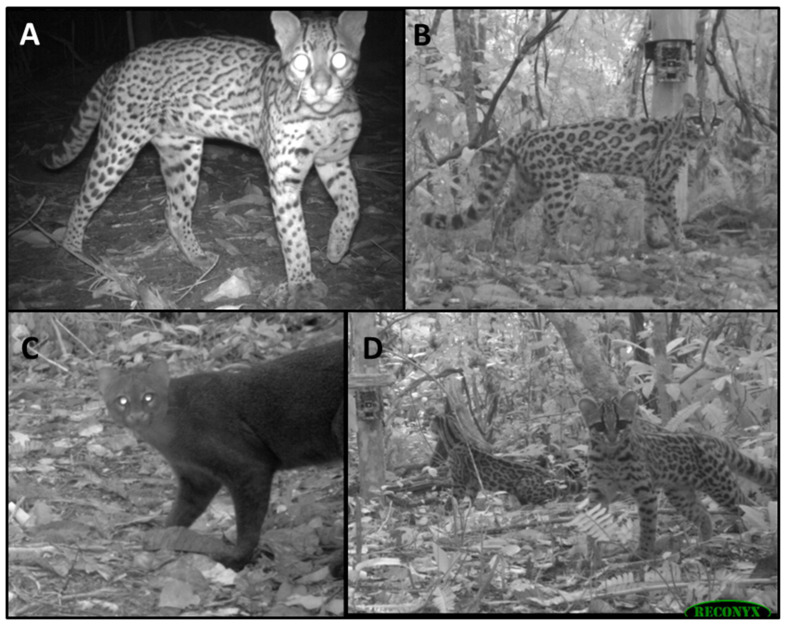
Examples of camera captures of the four focal species: (**A**) ocelot (*Leopardus pardalis*), (**B**) margay (*Leopardus wiedii*), (**C**) jaguarundi (*Herpailurus yagouaroundi*), (**D**) oncilla (*Leopardus tigrinus*). Note that cameras were attached approximately 30–40 cm above ground. The three species of spotted cat can be distinguished by general size and the relative length of the tail and limbs [11,12]. Ocelots are larger, about twice the size of oncillas or margays, with a short, thin tail relative to their body size. Margays are slenderer, with rounder heads, larger eyes, longer limbs, and a tail length about two-thirds to three-quarters of the length of the body. Oncillas are about the size of a small house cat, with very round heads, short ears, and shorter limbs and tail compared to margays.

**Figure 2 animals-14-00697-f002:**
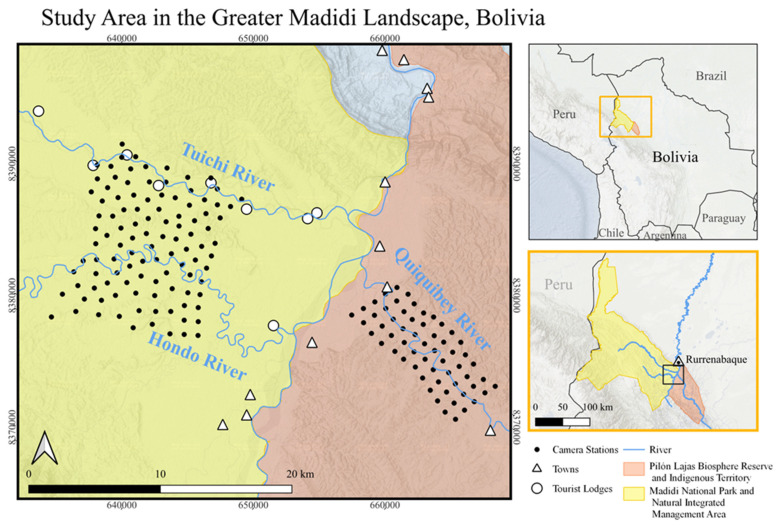
Map of the study site, showing camera stations along the three focal rivers of Quiquibey (survey 1), Hondo (survey 2), and Tuichi (survey 3). The inset maps show (**top**) the location of the Greater Madidi Landscape in northwestern Bolivia, and (**bottom**) the location of the study site within the Greater Madidi Landscape. Data for national park from UNEP-WCMC.

**Figure 3 animals-14-00697-f003:**
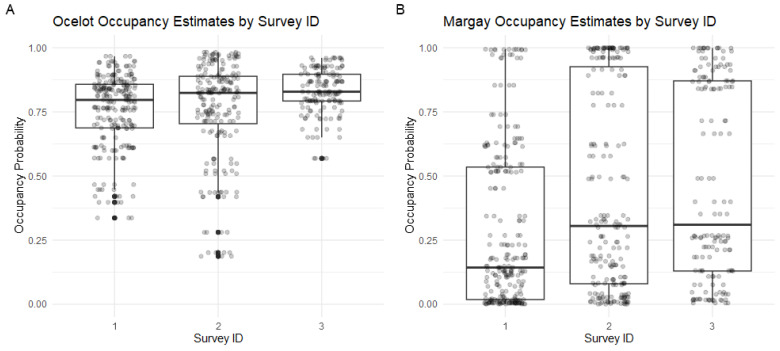
Occupancy estimates for camera stations in each of the three surveys for ocelots (**A**) and margays (**B**). Ocelot occupancy is relatively uniform in surveys 1 and 2, but significantly higher in survey 3. Margay occupancy is lower than ocelot across surveys, but appears lowest in survey 1 (though not statistically different).

**Figure 4 animals-14-00697-f004:**
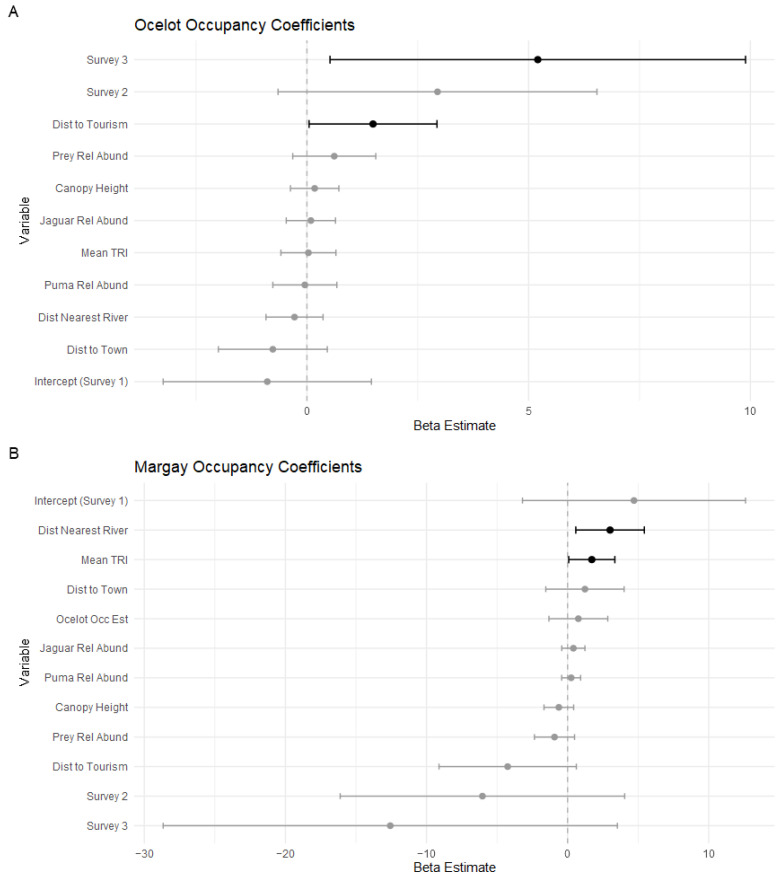
Coefficient plots from the single-species occupancy models for ocelots (**A**) and margays (**B**). The circle indicates the coefficient β-estimate, with bars showing the 95% Confidence Interval. Non-significant variables are gray, while significant variables are black.

**Figure 5 animals-14-00697-f005:**
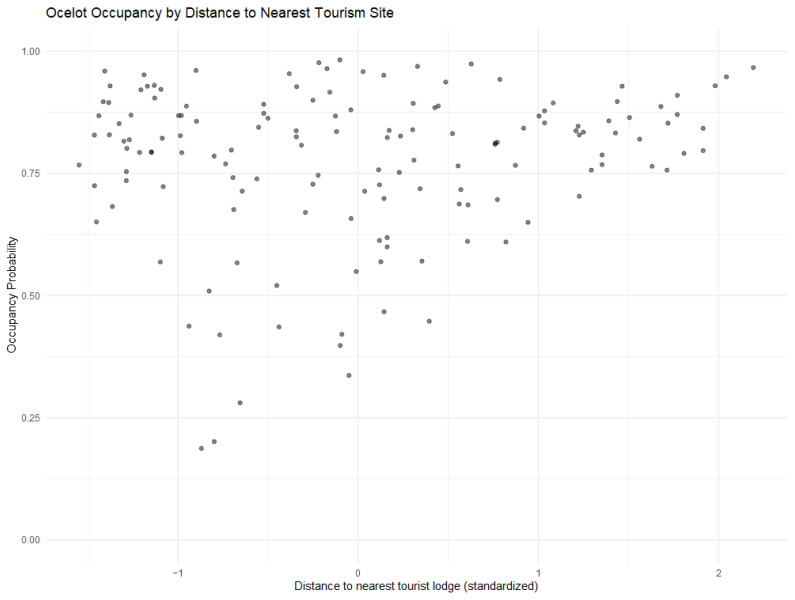
Ocelot occupancy estimates at each camera station by distance of that station to the nearest tourist lodge/site (standardized values). Our model found a statistically significant positive association between occupancy and distance from tourism sites, though note that the data shows a generally high occupancy throughout.

**Figure 6 animals-14-00697-f006:**
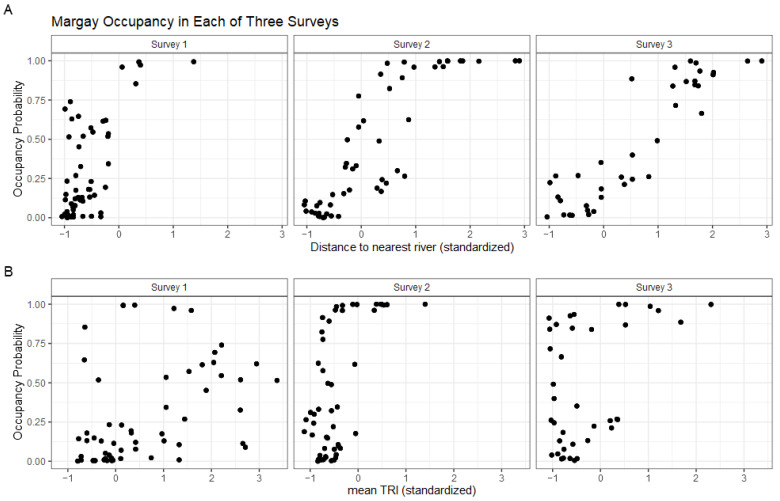
Margay occupancy estimates at each camera station, divided by the three surveys, by (**A**) distance to the nearest river and (**B**) mean TRI (terrain ruggedness index), with both variables standardized. The model showed a significantly positive association between margay occupancy and both variables, which does not appear to be a survey-specific relationship.

**Table 1 animals-14-00697-t001:** Summary of occupancy covariates calculated for each camera station, to be used in species occupancy models. Ocelot prey was considered any animal <15 kg. Margay/Jaguarundi/Oncilla prey was considered any animal <1 kg.

Type	Covariate	Mean (Minimum, Maximum)
Human Variables	Distance to nearest road (m)	18,809 (7176, 30,211)
	Distance to nearest town (m)	10,947 (629, 19,494)
	Distance to nearest tourist lodge (m)	7435 (331, 17,489)
Non-Human Variables	Mean TRI ^1^ (m)	25.13 (10.00, 70.53)
	Canopy Height (m)	26.23 (0, 32)
Relative Abundance Indices	Jaguar (obs/100 days)	1.547 (0, 13.043)
	Puma (obs/100 days)	0.521 (0, 13.043)
	Ocelot Prey (obs/100 days)	117.5 (2.12, 476.60)
	Margay/Jaguarundi/Oncilla Prey (obs/100 days)	27.3 (0, 145.65)

^1^ Terrain Ruggedness Index [61].

**Table 2 animals-14-00697-t002:** Mean values of occupancy probability ($\hat{\Psi }$ ± SE) and detection probability ($\hat{p}\pm SE$) using the single-species occupancy model for each species. Detection/non-detection is reported for each survey as the number of camera stations with an observation of that species/the number of stations without and the percentage of stations that did observe the species. The number of independent observations (non-consecutive or consecutive but >30 min apart) for each survey and overall are given. The jaguarundi model was poorly fit due to low detections, so only the detection/non-detection and number of independent observations are reported. Oncillas were only detected during the first survey.

Species	Survey	Occupancy Prob. ($\hat{\Psi }$) ± SE	Detection Prob. ($\hat{p}$) ± SE	Detection/Non-Detection (% with Detection)	Number of Observations
Ocelot	1	0.7574 ± 0.1310		42/17 (71.2%)	215
	2	0.7571 ± 0.1310		36/16 (69.2%)	186
	3	0.8308 ± 0.1168		30/8 (78.9%)	77
	Overall	0.7760 ± 0.1274	0.4711 ± 0.02753	108/41 (72.5%)	478
Margay	1	0.2902 ± 0.1739		8/51 (13.6%)	14
	2	0.4402 ± 0.1387		11/41 (21.1%)	18
	3	0.4740 ± 0.1774		9/31 (23.7%)	15
	Overall	0.3894 ± 0.1625	0.1466 ± 0.0310	28/121 (18.8%)	47
Jaguarundi	1			7/52 (11.9%)	12
	2			4/48 (7.7%)	5
	3			2/36 (5.3%)	2
	Overall			13/136 (8.7%)	19
Oncilla	Overall (1)			2/57 (3.4%)	5

## Data Availability

Data are available upon request. Data do not reveal coordinate locations of individual animals. For information, contact Courtney Anderson (anderson.4177@osu.edu).

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
