# Peer review of "Small Felids Coexist in Mixed-Use Landscape in the Bolivian Amazon"

_animals, 2024, doi:10.3390/ani14050697_

Round 1

Reviewer 1 Report

Comments and Suggestions for Authors

Review of animals-2802352

This is a very well-organized paper aiming to understand occupancy interactions of four syntopic small felids in Bolivia. The study is well developed with a robust methodological approach and data analysis methods. Interestingly, the authors found feeble evidence of small felid occupancy being affected by human activity at variable degrees, and no support for the intraguild suppression of smaller species by larger cats.

The introduction flows very well and clearly presents the aims of the study in an orderly manner and states the predictions that will be tested.

The methods are very well-described.

There is only one problem with the study period, where the authors need to provide an explanation (either ecological or technical) why the study was limited only between June and November. If the period was spread throughout the year, also including the wet season, would this change some of the results?

The other question would deal with the real syntopy of the different studied species. Why did not the authors include the occupancy of other species to test the occupancy of a give species? This could eventually provide some real evidence between potential mutual competition. It would be nice if the authors devote a few words on that in the methods section and in the discussion.

Results are very clear and explicitly presented.

The discussion explains very well and analyzes the results and tries to provide rebuts ecological explanations of the findings. Moreover, the authors also tackle conservation issues, as these are main in the introduction.  

Reviewer 2 Report

Comments and Suggestions for Authors

It should be noted that the authors undertook research on the functioning of populations of four species of small wild cats in urbanized areas. This type of research, although conducted more and more often, is extremely important in terms of taking actions in the field of population management and their protection. Therefore, the manuscript deserves to be published. Nevertheless, I have some comments that I think should improve this study.

Line, 53-55, this should be placed at the end of the introduction as the purpose of the research and the entire study,

The line, 82-90, is more of a methodology than an introduction?

Line 90-95 is the research hypothesis and should be placed at the end of the introduction.

Generally speaking, the purpose of the work and the hypotheses should be organized so that they are clear and understandable to the recipient. Moreover, you need to put them together at the end of the introduction.

Another thing is that the material and methods chapter is too long, especially the part about statistical calculations. It is necessary to revise and shorten it, while providing the most important information so that the results can be freely interpreted.

Line 246-251, this description is vague and thus difficult to interpret?

The discussion chapter is arranged in a very strange way and, in my opinion, does not meet publishing standards, and above all, it presents the results (!). The results should be described in an earlier chapter, and the discussion is nothing more than their reference to the research of other authors in similar fields. This must be revised.

In the discussion, the authors emphasize that the only significant variable from the models performed was the distance from the nearest tourist area and it only concerned Ocelots. I don't know if the fact of easy prey in the form of waste in these areas was taken into account and it mainly affects the behavior of wild animals and in many cases, in a certain way, fear of humans and their loss, and at the same time preference for such areas (commensalism). After all, commensalism is mentioned as one of the basic reasons for the domestication of animals. It is worth the authors paying attention to this, especially in terms of the results obtained.

Taking into account the above comments, the work, although the results for some species are very scarce, in my opinion, should be published after a thorough revision and the introduction of suggested corrections.

Reviewer 3 Report

Comments and Suggestions for Authors

My task was easy with this manuscript. It was written with high proficiency, the chosen methods are suitable and well explained.

The results were carefully evaluated regarding to species with insufficient data.

 Authors are aware of the potential flaws of some of their results and they clearly evaluate instead of overlook them. I always appreciate this approach since the boundaries of the actual research are clear, and Authors emphasize what can and cannot be answered from the results.

I only have a handful of comments to the manuscript:

Simple summary

line 20: "... effective division of roles..." Can't we call it niche segregation?

Abstract:

line 24: "..., particularly in highly diverse, complex." The sentence sounds unfinished:  '....habitats.' ?

Results

line 273: "...(241 Calvin Klein, 234 Chanel), suggesting no preference by ocelot." That means that these scents were ineffective?

line 279-280: "...was negatively correlated with distance to tourist lodge, prey relative abundance, and canopy height, but none significantly..."

This and other similar inferences sound biased since these were non-significant, however it creates a feeling that Authors want to arbitrarily confirm their hypotheses even if it was not statistically proven.

This way one could say that jaguar, puma and ocelot relative abundance was positively associated with margay occurrence though not significant (when considering lines 282-283). On the other hand, maybe there was a latent variable that "pulled" these variables towards the positive side.

I wonder whether the significance of results would have changed if other methods e.g. a bayesian approach from the package spOccupancy had been used.

Discussion

 Some thoughts here rather belong to the results.

Round 2

Reviewer 2 Report

Comments and Suggestions for Authors I still think that the material and methods section is too long and the authors should shorten it. You can also improve the summary. I leave it to the decision of the managing editor. The rest is fine, so I recommend the manuscript for publication.
